# Availability of alternative prey rather than intraguild interactions determines the local abundance of two understudied and threatened small carnivore species

Alejandro Hernández-Sánchez[�], Antonio Santos-Moreno [ID]*[�]

Laboratorio de Ecología Animal, Centro Interdisciplinario de Investigación para el Desarrollo Integral Regional-Unidad Oaxaca, Instituto Politécnico Nacional, Oaxaca, México

উ These authors contributed equally to this work.
* asantosm90@hotmail.com

## Abstract

Intraguild interactions influence the structure and local dynamics of carnivore mammals' assemblages. The potential effects of these interactions are often determined by the body size of competing members and may result in negative relationships in their abundance and, ultimately, lead to species exclusion or coexistence. The relative importance of interspecific interactions along with landscape characteristics in determining population patterns of understudied and threatened sympatric small carnivores, such as skunks, remains poorly documented. Therefore, we assessed the spatiotemporal variation in the abundance of American hog-nosed skunks *Conepatus leuconotus* and pygmy spotted skunks *Spilogale pygmaea* and the effect of interspecific interactions, resource availability, and habitat complexity on their local abundance in areas with the deciduous tropical forest south of the Mexican Pacific slope. We used presence-absence data for skunk species from three camera-trapping surveys between 2018 and 2020 in combination with Royle-Nichols occupancy models fitted in a Bayesian framework to estimate abundance, incorporating the effects of covariates related to the factors evaluated. We analyzed the relationship between the abundances of skunks using Bayesian Generalized Linear Models. Both skunk species showed significant differences in their abundances between seasons and between study sites. Overall, pygmy skunks were more abundant than hog-nosed skunks. We found negative relationships among the relative abundances of skunks during the dry seasons, but no evidence that local abundance is governed by the competitive dominance of the larger species. Patterns of skunk abundance were better explained by prey availability and other predictors related to habitat complexity, rather than interspecific interactions, since these models showed the highest predictive accuracies and strong positive and negative relationships. Our study highlights the underlying factors that determine the local abundance of these understudied and threatened small carnivores, allowing us to better understand the mechanisms that govern their coexistence for effective management and conservation of species in seasonal environments.

**Data Availability Statement:** All relevant data are within the manuscript and its Supporting Information files.

**Funding:** The Consejo Nacional de Ciencia y Tecnología of Mexico awarded a grant (#593502) for graduate studies to A.H.S., and the Instituto Politécnico Nacional of Mexico provided financial support (#SIP-20180613, SIP-20196209 and SIP-20200030) to carry out the project to A.S.M.. The funders had no role in study design, data collection and analysis, decision to publish, or preparation of the manuscript.

**Competing interests:** The authors have declared that no competing interests exist.

# Introduction

Mammalian carnivores play a key role in structuring and local dynamics of ecological communities in terrestrial systems, since these species not only regulate prey populations but may also affect other members of the group through top-down effects [1–3]. Therefore, carnivores may act as competitors and predators of one another at the same trophic level [1, 4, 5], meaning that intraguild interactions have the potential to shape carnivore assemblages [6–8].

Intraguild interactions can result in exploitative competition when species compete indirectly for the use of shared resources or interference competition in which one species is directly agonistic towards another through a set of behaviors ranging from defensive displays passive until aggression or interspecific killing [2, 4, 9]. These competitive interactions often occur between carnivores similar in terms of body size, diet, habitat, and activity pattern [1, 5]. The intensity of interference competition, in particular, is strongly determined by the body size of the interacting species, with the smallest member almost invariably occupying the subordinate position [2, 6, 7], and may be driven by diet overlap during periods of food scarcity [6, 10]. Additionally, the strength of competition is more intense between same-family species pairs at intermediate and large differences in body size [7].

Large carnivores can suppress populations of medium-size carnivores or mesocarnivores, and this, in turn, suppress smaller carnivores' populations through resource competition, intraguild predation or interspecific killing, and fear-driven spillovers [3, 6, 8, 11]. The effects of these interactions result in inverse relationships between the abundances of competing carnivores [2, 11, 12], with reduced densities of subordinate species [13–17], and may lead ultimately to competitive exclusion or species coexistence [2–4]. For example, some canids show a clear negative relationship in their relative abundances in regions of North America [13, 17, 18]. Densities observed in small neotropical felids (< 8.0 kg) are also lower in areas of South America where the guild's mid-sized member, ocelot *Leopardus pardalis*, is abundant [15, 19]. The effects of interference competition, however, decrease when the dominant competitor is found in small numbers and, consequently, subordinate carnivores reach high densities [14, 15, 20, 21]. Nevertheless, the abundance of potential competitors could also reflect associations with food availability or differences in habitat preference and show weak evidence of competitive interactions [16, 22, 23].

Although agonistic encounters between small carnivores have been considered relatively insignificant [5], some studies have documented interspecific interactions involving members of the family Mephitidae [24–26]. In addition, skunks are under notably high potential predation pressure [5, 8]. The American hog-nosed skunk *Conepatus leuconotus* (1.1–4.5 kg; hereafter hog-nosed skunks) and the pygmy spotted skunk *Spilogale pygmaea* (0.1–0.3 kg; hereafter pygmy skunks) overlap in their ranges within the Mexican Pacific slope [27, 28]. Both species feed mainly on insects and some small vertebrates when insect availability is low [29, 30], and are found in habitats with vegetation cover, although may also use open areas [30–32]. They inhabit the deciduous tropical forest in this region [27, 33], an ecosystem with marked environmental seasonality and temporal changes in resource availability and vegetation structure [34–36]. In this regard, the hog-nosed and pygmy skunks share similar ecological attributes that may predispose them to intraguild interactions in a seasonal environment with periods of resource scarcity, so this natural system provides the opportunity to investigate the possible effects of interspecific competition on the abundance of sympatric species. Likely, intraguild dynamics are also influenced by mesocarnivores presence (e.g., ocelots, coyotes *Canis latrans*), which could act as top predators in some areas of the Mexican Pacific slope where large carnivores are absent [37], affecting negatively populations of skunks through predation.

To date, scarce data are available on the abundance of *C. leuconotus* and *S. pygmaea* in Mexico, despite their populations being in decline [38, 39]. Some research has recorded inverse temporal variations in density between hog-nosed skunks and hooded skunks *Mephitis macroura* in seasonal tropical habitats [32, 40], and suggests that the largest-sized species determines the dynamics of interactions when it presents a high relative abundance [41]. In other assemblages, however, there is also evidence that the subordinate skunk could have some competition dominance by being in higher numbers [25]. Despite the above, the relative importance of interspecific interactions and landscape characteristics remains poorly documented in determining skunks' abundance patterns. Data provided by camera traps in combination with novel hierarchical modeling approaches allow us to estimate the species abundance and increase the ecological information of unmarked animal populations [23, 42]. The Royle-Nichols model (hereafter R-N) is a suitable alternative for estimating population size from presence-absence data, accounting for imperfect detection and incorporating covariate effects to avoid biased estimates [43, 44].

Knowledge of the underlying ecological factors that affect the abundance of small carnivores will allow us to understand the mechanisms that govern the coexistence of species for effective management and conservation of understudied and threatened skunks in seasonal environments. The goals of our study were to assess the abundance and spatiotemporal variation of hog-nosed and pygmy skunks, and to assess the effect of interspecific interactions, resource availability, and habitat complexity on their local abundance in areas with deciduous tropical forest south of the Mexican Pacific slope. Based on the effect of body size and taxonomic relationship on competitive interactions in carnivores [6, 7] and the intraguild dynamics between skunks in similar habitats [41], we hypothesized that hog-nosed skunks (larger species) would be more abundant than pygmy skunks, predicting a negative relationship in their abundance that may vary with population changes of the dominant competitor. We also expected intraguild predation to hurt the local abundance of both species due to their potentially high predation risk [5]. The effects of these interactions may be evident during the dry season when they are more likely to occur by resource scarcity [6, 7]. Alternatively, we hypothesized that resource availability and habitat structure would be more influential than intraguild interactions on the abundance of skunks.

## Materials and methods

### Study area

The study was carried out in the Protected Natural Area Huatulco National Park (15˚39'12"N to 15˚47'10"N and 96˚06'30"W to 96˚15'00"W), located in Santa María Huatulco municipality on the coast of the Oaxaca state, south of the Mexican Pacific slope (Fig 1A). The Huatulco National Park has 6,374.98 ha of land area [45], and is part of the Priority Terrestrial Region Sierra Sur-Costa de Oaxaca (PTR-129 [46]). The climate is warm subhumid with the lowest humidity, characterized by strong seasonality [47]. The average annual temperature fluctuates between 26–28˚C, and total annual precipitation varies between 800–1,200 mm, with the rainy season occurring from June to October and the dry season from November to May [45, 47]. The dominant vegetation is the deciduous tropical forest [48], which presents natural elements that stand out nationally and internationally for their conservation [45]. This protected area harbors one of the last well-conserved fragments of this vegetation [45]. It also shelters native and exotic mesopredators such as coyotes, ocelots, and feral dogs *C. lupus familiaris* [37, 49, 50]. Still, it is virtually free of top predators as the cougar *Puma concolor* has not been recorded for more than ten years [49].

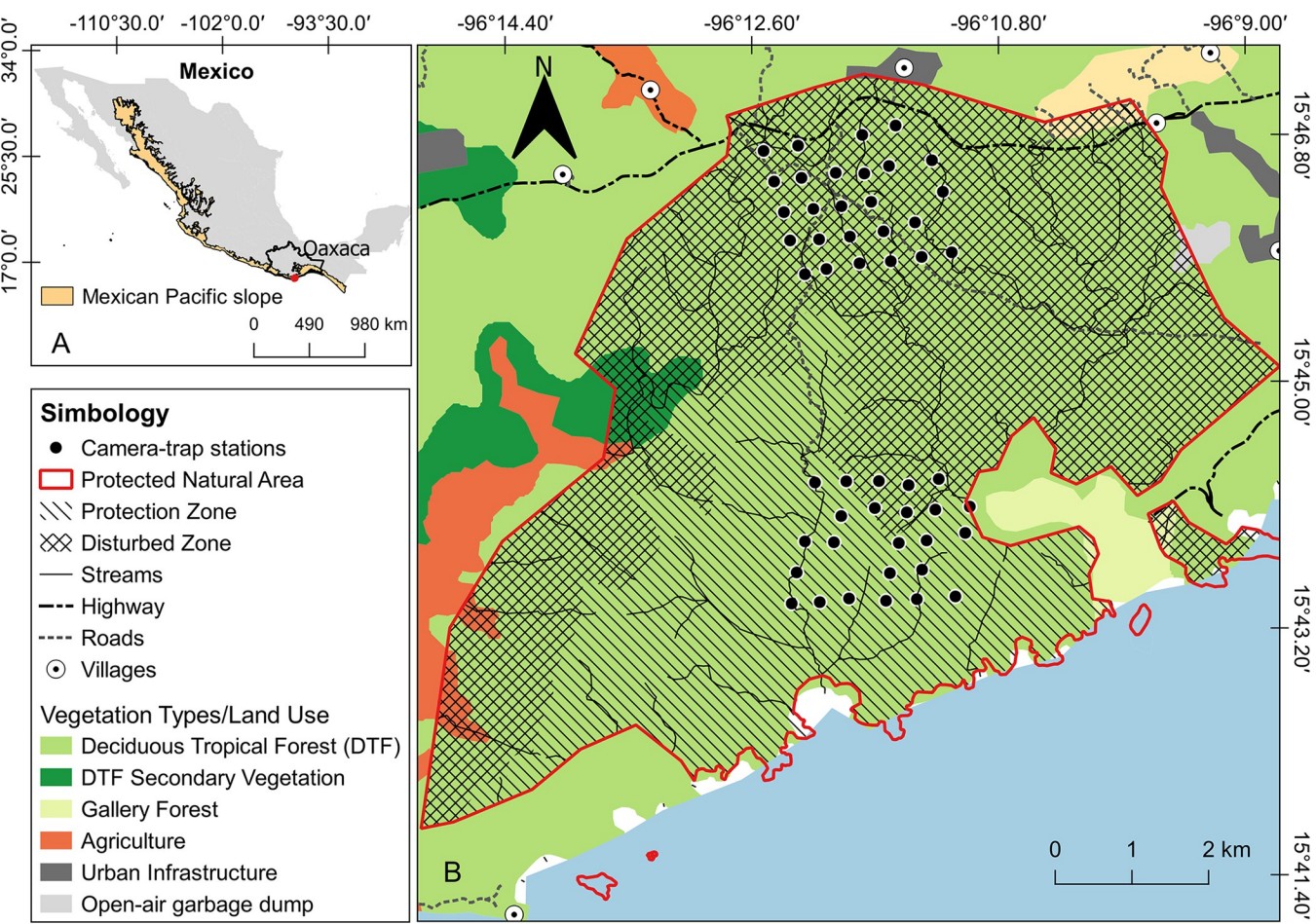

**Fig 1. The geographic location of the study area.** Republished from [51] under a CC BY license, with permission from [Juan Morrone], original copyright [2017]. A) Location of the Huatulco National Park on the coast of Oaxaca state, south of the Mexican Pacific slope. B) Location of the camera-trap stations in the protected zone and disturbed zone inside the deciduous tropical forest at the Protected Natural Area. This figure was prepared using spatial datasets available for free download online (see methods).

The location and spatial characteristics of the study area are shown in Fig 1 (created for illustrative purposes by the first author, AHS). The spatial datasets used were freely downloaded online from official websites, as indicated below: state political division, land use and vegetation, hydrographic network, and roads national network generated by the National Institute of Statistics and Geography of Mexico (https://www.inegi.org.mx/temas/), Mexican Pacific slope [51], and Huatulco National Park [52]. The spatial datasets were projected using QGIS 3.4.6 software [53].

## Definition of sampling sites

We defined two sampling sites based on the zoning of the Huatulco National Park, one in the protected zone and the other between areas of restricted use, sustainable harvesting, and recovery (hereafter disturbed zone). A detailed description of these zones was made by Hernández-Sánchez and Santos-Moreno [54]. The sampling sites have a similar floristic composition and are separated by a linear distance of 3 km (distance between the nearest camera trap stations). We considered the sampling sites to be spatially independent because the separation distance far exceeded the average male home range size of hog-nosed skunks (1.94 km$^2$ [55]).

## Camera trap survey

We conducted a systematic survey using camera traps from November 2018 to October 2020 to record the presence or absence of skunk species in both, the protected and disturbed zones in the study area. We installed 30 camera trapping stations in each zone, designing a grid of 6 by 5 stations regularly spaced 430 m apart (Fig 1B). However, only 48 stations functioned totally because several camera traps were stolen during the study. The sampling design was defined based on estimates of the home range of pygmy skunks (0.20 km² [56]) and the population densities of hog-nosed skunks in similar environments (0.5−1.3 individuals/km² [32, 40]). All sampling stations consisted of an unbaited camera trap, which was placed on trees approximately 20−30 cm above the ground inside the forest. We used four camera trap models: Bushnell Trophy Cam® and Bushnell Trophy Cam w/Viewscreen® (Bushnell Outdoor Products, Overland Park, Kansas), ERE-E1 (EREAGLE Technology Co., Ltd, Shenzhen, China), and Moultrie 990i Digital Game Camera (Moultrie Products, LLC, Birmingham, Alabama). Camera traps were configured to take a three-photo burst and record one 10−15 s video for each trigger event (with a 10 s delay between consecutive events), to remain active 24 h every day, and to record the date and time on all photos and videos. Data organization and independent records processing for target species were performed using the R package *camtrapR* [57]. We considered two consecutive photographs of hog-nosed or pygmy skunks captured 24-h period apart as independent detections in each sampling station [23, 58].

## Predictive covariates

Based on previous ecological research on the focal and similar species [25, 32, 41, 55], we compiled information on covariates that could affect their abundance in the study area. We selected a set of biologically important predictor variables to model the abundance of each skunk, including four variables related to interspecific interactions, four to resource availability, and two to habitat complexity (Table 1). Covariate values were recorded at each sampling station using field methods and remote sensing from geographic information systems. Vegetation structure measurements were taken in 20 by 20 m sampling quadrants centered on the location of each sampling station and in 4 by 4 m sub-quadrants located at the corners and center of each quadrant [59, 60]. Additionally, we selected two covariates for modeling the detection probability of skunks (Table 1).

All continuous covariates were standardized (mean = 0; standard deviation = 1) before analysis to allow for improved parameter estimation and facilitate the comparison of model estimates and interpretation of relative effect sizes [67]. We tested multicollinearity between covariates using the variance inflation factor (VIF) in the R package *HH* 3.1−49 [68]. We considered evidence of collinearity with VIF values > 5 [69, 70], and in this case, we excluded highly collinear predictors from the same models (S1 Table).

## Modeling framework

We used the R-N model to estimate the abundance of skunk species from detection-non-detection data [43]. The R-N model considers the heterogeneity in site-specific detectability to be derived from variation in local abundance, i.e., it is an occupancy model of abundance-induced heterogeneity [43, 44, 67]. The relationship between heterogeneous detection probability and abundance is $P_{ij} = 1 − (1 − r_j)^{N_i}$, where $p_{ij}$ is the probability of detecting the species at site i, $r_j$ is the probability of detecting an individual, and $N_i$ is the number of individuals at site i [43, 44]. In this way, the R-N model provides estimators of the parameters λ and r, defined as the average abundance per site and the detection probability, respectively [43].

**Table 1. Selected biologically important covariates and their predicted effects on modeling two skunk species' abundance and detection probability.**

| Covariates | Abbreviation | Description | Mean (SD) | Prediction |
|---|---|---|---|---|
| **Abundance** | | | | |
| *Interspecific interactions* | | | | |
| Presence of competitors | Competitor | Number of independent records (> 1 h apart) of American hog-nosed skunks (*Conepatus leuconotus*) and pygmy spotted skunks (*Spilogale pygmaea*) per site | 0.25 (0.45) and 4.46 (8.66) detections | Abundance of the subordinate competitor will decrease with the presence of the dominant competitor |
| Presence of coyotes | Coyotes | Number of independent records (> 1 h apart) of coyotes (*Canis latrans*) per site | 0.91 (1.96) detections | Abundance will decrease with the presence of coyotes, a potential predator of skunks |
| Presence of dogs | Dogs | Number of independent records (> 1 h apart) of dogs (*Canis lupus familiaris*) per site | 0.51 (1.36) detections | Abundance will decrease with the presence of dogs, a potential exotic predator of skunks |
| Presence of ocelots | Ocelots | Number of independent records (> 1 h apart) of ocelots (*Leopardus pardalis*) per site | 1.92 (7.05) detections | Abundance will decrease with the presence of ocelots, a potential predator of skunks |
| *Resource availability/ Habitat complexity* | | | | |
| Potential burrows | Burrows | Number of potential burrows in each sampling quadrant with an entrance $\geq$ 5 cm in diameter for pygmy skunks and $\geq$ 15 cm for hog-nosed skunks (obs. pers., [61]) | 16.04 (8.60) and 2.77 (3.33) burrows | Abundance will be greater in sites with a greater number of potential burrows. |
| Availability of small mammals | Avasmam | Relationship between the number of independent records (> 1 h apart) of small mammals (rodents and marsupials < 1 kg in weight) and the sampling effort for each camera-trap station multiplied by 100 | 4.96 (6.72) records / 100 trap-nights | Abundance will increase with higher availability of small mammals, considered as potential prey [29, 30] |
| Soil humidity (as a proxy for litter arthropod availability) | Soilhum | Average Modified Normalized Difference Water Index (MNDWI) values in a 200 m radius circular buffer around each sampling station, which were estimated from atmospherically corrected Landsat 8 satellite images with QGIS 3.4.6 software [53] Satellite images were downloaded from the United States Geological Survey (https://www.usgs.gov) Values range from -1 (no presence of water) to 1 (higher content or presence of water) [62] | -0.58 (0.02) | Abundance will be greater in sites with higher soil humidity, due to its direct relationship with the diversity and abundance of insects [35] |
| Distance to the nearest water source | Dishwater | Euclidean distance from the camera trap stations to the nearest water source (stream, pond, or waterhole) in the study area The distances were estimated based on the MNDVI, which also allows the mapping of water bodies [62] | 3.77 (2.32) km | Abundance will be higher closer to water sources |
| Shrub cover | Shrcover | Estimation of the shrub cover from the maximum ($d_1$) and perpendicular ($d_2$) lengths of the shrub crowns using the formula: SC = $\Sigma$ ($\pi$ * (1/4 [$d_1$ + $d_2$]) $^2$) Measurements were taken in the five sampling sub-quadrants | 26.55 (21.36) m$^2$ | Abundance will be higher in sites with higher shrub cover. |
| Canopy cover | Cancover | Average Normalized Difference Vegetation Index (NDVI) values in a 200 m radius circular buffer around each sampling station from corrected Landsat 8 satellite images Values range from 0 (non-forest) to 1 (dense forest cover) | 0.61 (0.21) | Abundance will be higher at sites with higher canopy cover |
| **Detection** | | | | |
| Sampling effort | Effort | Number of nights each camera-trap station was active | 89.55 (31.29) trap-nights | Higher sampling effort values will increase the detection rate of skunks |
| Lunar illumination | Lunillu | Average values for each sampling occasion of the illuminated fraction of the visible lunar surface got with the R package suncalc 0.5.1, varying from 0.0 (new moon) to 1.0 (full moon) [63] The fraction is computed using astronomical algorithms by Meeus [64] based on a reference date, time, and location (time zone) [63] The calculations do not incorporate the effects of vegetation or cloud cover on lunar illumination | 0.49 (0.35) | Brighter nights should increase the detection rate of hog-nosed skunks or decrease that of pygmy skunks, consistent with reports from similar species [65, 66] |

**Table 2. Details of the three surveyed seasons in the study zones and the number of detections of the two skunk species at Huatulco National Park, Oaxaca, Mexico.**

| Surveyed season | Study zone | Surveyed period | Number of camera-trap stations | Trap-nights | Number of detections | |
|---|---|---|---|---|---|---|
| | | | | | American hog-nosed skunk | Pygmy spotted skunk |
| Dry season 2019 | Disturbed zone | Dec 1, 2018-Mar 30, 2019 | 26 | 2,585 | 10 | 106 |
| | Protected zone | Dec 13, 2018-Apr 11, 2019 | 21 | 1,613 | 0 | 30 |
| Rainy season 2019 | Disturbed zone | Jun 1, 2019- Sep 28, 2019 | 26 | 2,836 | 24 | 121 |
| | Protected zone | Jun 13, 2019- Oct 10, 2019 | 22 | 1,868 | 3 | 62 |
| Dry season 2020 | Disturbed zone | Nov 19, 2019-Mar 17, 2020 | 24 | 1,795 | 7 | 62 |

The R-N model assumes that the population is demographically closed (the population size must not change during the study period), the detection of an animal at a site is independent of the detection of any other animal, and the detection probability is equal for all individuals [43, 44]. Because sampling was carried out over two years, we truncated sampling to three surveys per study zone corresponding to the yielded climatic seasons: dry season 2019, rainy season 2019, and dry season 2020 (Table 2; S1 Dataset). Each survey consisted of 120 consecutive trap nights to meet population closure among repeated surveys, considering the breeding and non-breeding (when offspring become independent) seasons for both skunk species [30, 31]. For modeling, we pooled the detection records of each species for each camera-trap station (analysis site) across 12-night sampling occasions. So the detection matrices consisted of 21–26 sites (depending on the survey, see Table 2) and 10 sampling occasions, where 1 denotes that the target species was detected at least one site during a given sampling occasion and 0 that it was not detected.

The R-N model requires spatial independence of the sampling stations, meaning that the camera traps must be far enough apart that they do not detect the same individuals [42]. A sampling design with a spatial resolution between camera traps close to the target species' home range may allow this model to provide reliable estimates of absolute abundance [71–73] so we have interpreted a site's abundance in absolute terms for pygmy skunks. By contrast, since the distance between sampling stations was less than the home range diameter of hog-nosed skunks, we suspected a violation of the assumption of independence among sites and considered abundance per site for this species as relative, this is, the number of individuals using a site during a given time [67]. Other authors suggest this interpretation when the assumptions of site-structured models are not met [42, 67]. However, we accounted for possible spatial autocorrelation between nearby sites in the modeling framework, as shown below.

On the other hand, the R-N model usually estimates the average abundance ($\lambda$) with a positive bias of 10–22% when the detection probability of the species is low ($r \leq 0.1$) and the sample size is small (few sites sampled) [43, 44, 71, 74]. Nevertheless, this model produces unbiased estimates of the parameter $\lambda$ at sample sizes $\leq 100$ sites under some particular circumstances, with $\geq 10$ sampling occasions for low values of r or when $r \geq 0.2$ for $\geq 5$ sampling occasions [43, 71, 75]. The R-N model further performs reasonably well when dealing with species that have low densities or are territorial [67, 72, 74, 76]. Given that the target skunk species are considered rare, which translates into low densities [38, 39], the number of sites and sampling occasions in our modeling framework provide conditions favorable to estimate $\lambda$ from this robust approach.

We followed a two-stage modeling framework for building the candidate models of each skunk species. We first identified the best model for the detection probability while holding the average abundance per site constant. We modeled the detection probability without covariates (null model) and as a function of sampling effort and lunar illumination, one model with each covariate individually and another with both. We then built the models for the average abundance fixing the best-supported detection model. By our hypotheses, we designed three subsets of models to explain the abundance of species by variables related to (a) interspecific interactions, (b) resource availability and habitat complexity, or (c) a combination of all. We fitted biologically plausible models for each subset, including the individual and combined effects of the covariates shown in Table 1. The structure of the models was similar for both species to facilitate their comparison. Additionally, in the best-supported abundance models (non-spatial models) for hog-nosed skunks, we incorporated the spatial random effect parameter into the state process to account for the autocorrelation structure (spatial models). Spatial random effects were specified using the Restricted Spatial Regression (RSR) method extended to single-season occupancy models [77] since it is statistically (the random effect is not correlated with the fixed covariates) and computationally (the estimation is less intensive) more efficient [78].

We ran all models with the R package *ubms* 1.1.0 [79], in which R-N models are fit in a Bayesian framework using the programming language Stan [80]. Fitting models in the Bayesian framework is recommended when datasets have small sample sizes, few detections, and low detection probabilities [67, 81], as is the case for hog-nosed skunks. We used the default weakly informative priors for abundance parameters (intercepts and regression coefficients) and detection parameters [79, 82]. Specifically, we fitted the spatial models by including a call to the RSR function in the corresponding R-N models. We set the coordinate vectors, the distance threshold below which two sites were considered potentially correlated with each other (720 m, according to the average home range of hog-nosed skunks [55]), and the number of eigenvectors used when calculating the spatial random effect (recommended default value is 10% of the number of sites [78]). We ran the Bayesian R-N models using three Markov chain Monte Carlo (MCMC) chains of 2,000 iterations each, with a burn-in of 1,000 iterations per chain. Since *ubms* uses Stan's modeling language, a low number of iterations are required to reach model convergence and obtain stable parameter estimates [80, 83]. We assessed model convergence by checking that the $\hat{R}$ (Rhat) diagnostic statistic was less than 1.1 for each parameter and visually examining the traceplots [82, 84]. We also assessed the quality of MCMC sampling by verifying that the effective sample size (n_eff) was higher than 300 for all parameters [79].

We performed the selection of candidate models within and between subsets of each species using leave-one-out cross-validation for pairwise model comparisons (LOO-CV; [85]) in the package *ubms* [79]. Candidate models were ranked in descending order of their expected log pointwise predictive density (elpd), which estimates the predictive accuracy of the models [85]. We calculated the differences in elpd between each model and the superior model (Δelpd), the standard errors of these differences (SE [Δelpd]), and the model weights, analogous to the Akaike Information Criterion weights [85]. We interpreted that the model with the largest elpd performed better and considered the top model to have more support than another model if the absolute difference in elpd was greater than the standard error of that difference [79, 85]. We assessed the fit of the top-ranked model by obtaining residuals separately for both processes (detection and abundance) based on the approach of Wright et al. [86]. We also checked the top-ranked model goodness-of-fit with the MacKenzie-Bailey Chi-square test for occupancy models [87], using posterior predictive checks [79, 82]. Bayesian p-values near 0.5

indicated that the model fits well [67, 79]. We estimated the 95% Bayesian credible intervals (BCIs) of the posterior probability distribution to determine the significance of covariate effects. We considered a covariate to have strong effect if the 95% BCIs of its coefficient did not overlap with zero [67]. Finally, we generated response curves (marginal effects plots) of the strongly supported covariates included in the top-ranked models for both skunks.

## Abundance analysis

We got the predicted average abundance values for each site of hog-nosed and pygmy skunks during the three surveyed seasons at the two study zones from the best-supported R-N models. We then used Generalized Linear Models (GLMs) in a Bayesian framework with Poisson and Negative Binomial error distribution, both with log link function, to evaluate the influence of surveyed seasons and study zones on differences in the average abundance per site for each skunk species and to analyze the relationship between the abundances of skunk species during the surveyed seasons in each study zone. The Negative Binomial distribution includes a dispersion parameter that allows it to explain more variability than the Poisson distribution [67, 88, 89] and can be used when it exists a violation of the assumption of data independence [76].

We fitted Bayesian GLMs to model average abundance per site with covariates using the stan_glm function in the R package rstanarm 2.21.3 [90, 91] via MCMC in Stan [80]. We used the default weakly informative priors for the intercept and coefficients in the models of both distributions and the auxiliary parameter (reciprocal dispersion) in the Negative Binomial models [90, 91]. We ran the GLMs using the four default MCMC chains with 2000 iterations each, half of which were discarded as warm-ups. We assessed model convergence and sampling quality by checking the $\hat{R}$ statistic (Rhat < 1.1), effective sample size (n_eff > 1,000), and Monte Carlo standard error (MCSE) [84, 90]. A low MCSE relative to the estimated posterior standard deviation is desirable for a higher number of effective samples [90]. In addition, we assessed the influence of the observations on the model posterior distribution by verifying the Pareto k diagnostic statistic, which estimates how influential the data points are [85, 92]. Highly influential observations have k values greater than 0.7, indicating model misspecification or outliers [85, 93]. So, we defined the k threshold equal to 0.7 (above which the observation is flagged) when the diagnostics revealed problems, calling the loo function of the package rstanarm [91]. This specification allows for the model re-fitting by leaving out problematic observations one by one and directly computing their elpd contributions [85, 93]. We selected the candidate Possion and Negative Binomial GLMs with the largest elpd and highest model weight based on LOO-CV [85] using the package rstanarm [91], similar to the procedure described in R-N modeling. We calculated the 95% BCIs of the posterior probability distribution of the parameters to determine the significance of covariates effects. All data analyses were performed using the statistical software R [94].

Considering that the number of animals estimated at a survey point cannot be used as a surrogate for animal density [73], we estimated the density of skunk species in the effective sampling area, which was calculated by summing the area of the polygon formed by the sampling stations plus a buffer with the area of half the spacing distance between stations. We estimate the density of skunk species using the formula $D = \lambda * R$ / effective sampling area, where D is the number of individuals / $km^2$, $\lambda$ is the average number of individuals predicted per site, and R is the number of sites sampled [43]. However, to estimate the overall density of hog-nosed skunks, we followed the calculation of Thorn et al. [75], dividing D by the average number of sites probably used by individuals in the study areas according to the distance defined in the spatial autocorrelation analysis.

## Results

### Abundance of skunks

Consistent with the best-ranked GLM for each skunk species (S2 Table), the average abundances per site of both were influenced by surveyed seasons and study zones (Table 3). The dry seasons of 2019 (β = 0.96, BCI = 0.72 to 1.19) and 2020 (β = 0.41, BCI = 0.09 to 0.73) and the rainy season of 2019 (β = 0.75, BCI = 0.47 to 1.01), as well as the disturbed zone (β = 0.96, BCI = 0.72 to 1.19), had a strong positive effect. The protected zone conversely had a strong negative effect on the abundance of this species (β = -1.41, BCI = -2.10 to -0.80) on hog-nosed skunk abundance. The dry seasons of 2019 (β = 0.38, BCI = 0.04 to 0.72) and 2020 (β = 0.63, BCI = 0.25 to 1.00) and the rainy season of 2019 (β = 1.20, BCI = 0.90 to 1.49), as did the disturbed zone (β = 0.37, BCI = 0.03 to 0.70), also showed a strong positive effect on pygmy skunk abundance. The protected zone had a negative effect but without strong support on the abundance of pygmy skunks (β = -0.18, BCI = -0.60 to 0.23).

Overall, the Negative Binomial GLMs showed greater predictive accuracy in explaining the relationship in abundance between skunk species during the surveyed seasons in the study zones (S2 Table). The top-ranked GLMs fitted the data well, with acceptable diagnostic statistics (S3 Table). The relationship between the average abundance per site of pygmy skunks and hog-nosed skunks was negative but without strong support at the dry seasons of 2019 (β = -0.04, BCI = -0.30 to 0.21) and 2020 (β = -0.08, BCI = -0.46 to 0.31) in the disturbed zone (Fig 2A and 2C). In contrast, the relationship in abundance among skunks was positive and strong at the rainy season of 2019 in the disturbed zone (β = 0.19, BCI = 0.04 to 0.34; Fig 2B) and protected zone (β = 0.70, BCI = 0.33 to 1.06; Fig 2E), but without strong support at the dry season 2019 in the protected zone (β = 1.22, BCI = -1.68 to 4.66; Fig 2D). However, the association during the rainy season 2019 in the protected zone should be tempered with caution since it is driven by the only point with a high abundance of both skunk species (Fig 2E). The density of hog-nosed skunks ranged from 0.14 ind/km$^2$ during the rainy season 2019 in the protected zone to 2.42 ind/km$^2$ during the dry season 2019 in the disturbed zone. The density of pygmy

**Table 3. Parameter estimates of the best-ranked Bayesian Generalized Linear Models explaining the effect of surveyed season and study zone on the abundance of skunk species.**

|  | Parameter | Mean | SD | 2.50% | 97.50% | MCSE | n_eff | Rhat |
|---|---|---|---|---|---|---|---|---|
| **American hog-nosed skunk** | Dry season 2019 | 0.963 | 0.12 | 0.724 | 1.186 | 0.002 | 4291 | 1 |
|  | Rainy season 2019 | 0.747 | 0.136 | 0.473 | 1.012 | 0.002 | 4469 | 1 |
|  | Dry season 2020 | 0.419 | 0.165 | 0.087 | 0.73 | 0.003 | 4256 | 1 |
|  | Disturbance Zone | 0.962 | 0.121 | 0.721 | 1.189 | 0.003 | 1657 | 1 |
|  | Protection Zone | -1.413 | 0.334 | -2.102 | -0.803 | 0.007 | 2569 | 1 |
|  | Mean_PPD | 1.439 | 0.156 | 1.143 | 1.756 | 0.003 | 3913 | 1 |
|  | Log-posterior | -163.227 | 1.416 | -166.804 | -161.465 | 0.033 | 1880 | 1 |
| **Pygmy spotted skunk** | Dry season 2019 | 0.38 | 0.173 | 0.038 | 0.717 | 0.003 | 3600 | 1 |
|  | Rainy season 2019 | 0.628 | 0.191 | 0.249 | 1.003 | 0.003 | 3754 | 1 |
|  | Dry season 2020 | 1.204 | 0.152 | 0.901 | 1.492 | 0.003 | 3170 | 1 |
|  | Disturbance Zone | 0.373 | 0.172 | 0.034 | 0.702 | 0.004 | 1546 | 1 |
|  | Protection Zone | -0.178 | 0.213 | -0.596 | 0.226 | 0.005 | 1732 | 1 |
|  | Reciprocal dispersion[a] | 3.248 | 0.971 | 1.816 | 5.551 | 0.016 | 3508 | 1 |
|  | Mean_PPD | 1.955 | 0.248 | 1.521 | 2.496 | 0.004 | 3747 | 1 |
|  | Log-posterior | -212.228 | 1.634 | -216.426 | -210.091 | 0.041 | 1620 | 1 |

SD, Standard Deviation; MCSE, Monte Carlo standard error; n_eff, effective sample size; Rhat, diagnostic statistic (< 1.1).

[a] Smaller values of the parameter indicate greater dispersion [91].

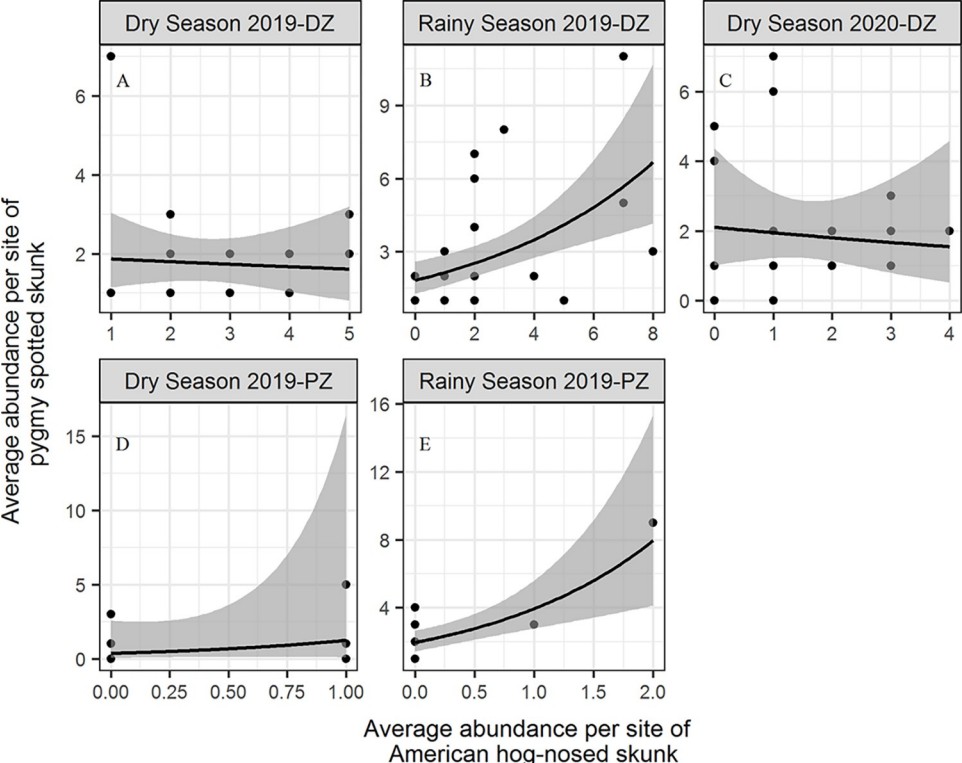

**Fig 2. Relationships in average abundance per site between skunk species during the surveyed seasons in the disturbed (DZ) and protected (PZ) zones at Huatulco National Park, Oaxaca, Mexico.** The bold lines indicate posterior means and the ribbons are 95% credible intervals.

skunks varied between 1.80 ind/km$^2$ during the dry season 2019 in the protected zone and 13.34 ind/km$^2$ during the rainy season 2019 in the disturbed zone.

## Factors affecting the detection and abundance of skunks

The best-ranked models with the highest predictive accuracy for the detection probability of hog-nosed skunks were the null model in the dry seasons of 2019 and 2020 and the one that included the sampling effort in the rainy season of 2019 (S4 Table). The full model had similar predictive accuracy to the top model in the rainy season (pairwise Δelpd was smaller than its SE[Δelpd]), but its weight was low (ω = 0.37). Meanwhile, the best-ranked models showing the highest predictive accuracy for the detection probability of pygmy skunks were the null model in the dry season of 2019 and the models that included lunar illumination and sampling effort in the rainy 2019 and dry 2020 seasons, respectively (S4 Table).

The candidate models that only included covariates related to resource availability and habitat complexity and the models that combined subsets of covariates showed the highest predictive accuracies and were the best supported in explaining the average abundance per site of both skunks as opposed to candidate models that only included interspecific interaction covariates (S5 Table and Table 4). The top-ranked model that explained hog-nosed skunk abundance during the dry and rainy seasons of 2019 had larger pairwise Δelpd than its SE[Δelpd], with a model weight equal to 1 (Table 4). All candidate models were similar in predictive accuracy during the dry season of 2020, but the top-ranked model had a much higher model weight than the others (ω = 0.72). The top-ranked models for this species did not improve their

**Table 4. Selection of best-ranked candidate Royle-Nichols models from model subsets explaining the abundance of skunk species at the each surveyed season using leave-one-out cross-validation for pairwise model comparisons.**

| | | Subset[a] | Model[b] | elpd | Δelpd | SE [Δelpd] | ω |
|---|---|---|---|---|---|---|---|
| **American hog-nosed skunk** | **Dry season 2019** | B | $r_{(.)}\lambda_{(diswater)}$ | -41.220 | 0.000 | 0.000 | 1.000 |
| | | C | $r_{(.)}\lambda_{(diswater + coyotes)}$ | -42.011 | -0.791 | 0.330 | 0.000 |
| | | Null | $r_{(.)}\lambda_{(.)}$ | -43.878 | -2.658 | 1.457 | 0.000 |
| | | A | $r_{(.)}\lambda_{(coyotes)}$ | -44.533 | -3.312 | 1.610 | 0.000 |
| | **Rainy season 2019** | C | $r_{(effort)}\lambda_{(cancover + avamam + soilhum + ocelots)}$ | -72.760 | 0.000 | 0.000 | 1.000 |
| | | B | $r_{(effort)}\lambda_{(cancover + avamam + soilhum)}$ | -73.593 | -0.833 | 0.664 | 0.000 |
| | | Null | $r_{(effort)}\lambda_{(.)}$ | -80.118 | -7.358 | 2.138 | 0.000 |
| | | A | $r_{(effort)}\lambda_{(ocelots)}$ | -80.569 | -7.809 | 2.181 | 0.000 |
| | **Dry season 2020** | C | $r_{(.)}\lambda_{(shrcover + coyotes)}$ | -29.581 | 0.000 | 0.000 | 0.715 |
| | | A | $r_{(.)}\lambda_{(coyotes)}$ | -30.002 | -0.421 | 1.053 | 0.000 |
| | | Null | $r_{(.)}\lambda_{(.)}$ | -30.120 | -0.539 | 1.706 | 0.285 |
| | | B | $r_{(.)}\lambda_{(shrcover)}$ | -30.229 | -0.649 | 1.612 | 0.000 |
| **Pygmy spotted skunk** | **Dry season 2019** | B | $r_{(.)}\lambda_{(avamam + diswater)}$ | -143.540 | 0.000 | 0.000 | 0.648 |
| | | C | $r_{(.)}\lambda_{(avamam + diswater + ocelots)}$ | -143.865 | -0.325 | 1.737 | 0.243 |
| | | A | $r_{(.)}\lambda_{(ocelots)}$ | -151.631 | -8.091 | 5.096 | 0.109 |
| | | Null | $r_{(.)}\lambda_{(.)}$ | -152.581 | -9.041 | 4.839 | 0.000 |
| | **Rainy season 2019** | C | $r_{(lunillu)}\lambda_{(avamam + diswater + coyotes)}$ | -193.257 | 0.000 | 0.000 | 0.702 |
| | | B | $r_{(lunillu)}\lambda_{(avamam + diswater)}$ | -193.933 | -0.676 | 2.118 | 0.298 |
| | | Null | $r_{(lunillu)}\lambda_{(.)}$ | -211.041 | -17.784 | 5.896 | 0.000 |
| | | A | $r_{(lunillu)}\lambda_{(coyotes)}$ | -211.149 | -17.892 | 6.009 | 0.000 |
| | **Dry season 2020** | B | $r_{(effort)}\lambda_{(shrcover + avamam)}$ | -72.616 | 0.000 | 0.000 | 1.000 |
| | | C | $r_{(effort)}\lambda_{(shrcover + avamam + coyotes)}$ | -73.298 | -0.682 | 0.591 | 0.000 |
| | | A | $r_{(effort)}\lambda_{(coyotes)}$ | -77.968 | -5.352 | 2.683 | 0.000 |
| | | Null | $r_{(effort)}\lambda_{(.)}$ | -79.545 | -6.929 | 2.247 | 0.000 |

elpd, expected log pointwise predictive density; Δelpd, pairwise differences in elpd (relative to the top model); SE[Δelpd], standard error of Δelpd; ω, model weight.

[a] Letters denote the set of covariates of the model evaluated for abundance: interspecific interactions (A), resource availability and habitat complexity (B), and a combination of both subsets (C).

[b] The abbreviations of the covariates in the candidate models are shown in Table 1.

predictive accuracy when incorporating spatial random effects to account for spatial autocorrelation across the three surveyed seasons (Table 5). The spatial random effect parameter showed high values and, therefore, it had lower statistical significance (S6 Table). Besides, the top-ranked model explaining pygmy skunk abundance during the dry season 2020 had larger pairwise Δelpd than its SE[Δelpd] and a model weight equal to 1, while the two best-supported models during the dry season 2019 and the rainy season 2019 showed similar predictive accuracy, although the top model had a higher model weight in both cases (ω > 0.65; Table 4). All of the top-ranked models showed acceptable fit based on posterior predictive checks, with Bayesian p-values from 0.55 to 0.68 for hog-nosed skunks and from 0.25 to 0.42 for pygmy skunks.

Detectability of hog-nosed skunks was positively related to sampling effort in the rainy season 2019 (β = 1.32, BCI = -0.06 to 2.81), although the relationship did not show strong support (Fig 3 and S7 Table). Meanwhile, the detectability of pygmy skunks strongly decreased with lunar illumination in the rainy season 2019 (β = -0.23, BCI = -0.46 to -0.01) and with sampling effort in the dry season 2020 (β = -0.69, BCI = -1.30 to -0.12, Fig 3 and S7 Table). The detection

**Table 5. Selection of non-spatial and spatial models explaining the abundance of hog-nosed skunks at the each surveyed season using leave-one-out cross-validation for pairwise model comparisons.**

|  | Model[a] | elpd | Δelpd | SE [Δelpd] | ω |
|---|---|---|---|---|---|
| **Dry season 2019** | Non-spatial | -41.157 | 0.000 | 0.000 | 1.000 |
|  | Spatial | -42.182 | -0.025 | 0.098 | 0.000 |
| **Rainy season 2019** | Non-spatial | -72.686 | 0.000 | 0.000 | 1.000 |
|  | Spatial | -73.071 | -0.386 | 0.362 | 0.000 |
| **Dry season 2020** | Non-spatial | -29.593 | 0.000 | 0.000 | 1.000 |
|  | Spatial | -29.862 | -0.269 | 0.068 | 0.000 |

elpd, expected log pointwise predictive density; Δelpd, pairwise differences in elpd (relative to the top model); SE[Δelpd], standard error of Δelpd; ω, model weight.

[a] Non-spatial models referred to the top-ranked abundance models in Table 4 and spatial models referred to those that additionally incorporated the spatial random effect parameter into the state process.

probability of hog-nosed skunks and pygmy skunks ranged from 0.02 to 0.09 and 0.11 to 0.20, respectively.

Distance to the nearest water source (β = 1.23, BCI = 0.28 to 2.42) in the dry season 2019 and the availability of small mammals (β = 0.88, BCI = 0.29 to 1.49) in the rainy season 2019 had a strong positive effect, while soil humidity (β = -1.29, BCI = -2.64 to -0.25) and canopy cover (β = -1.04, BCI = -1.74 to -0.43) in the rainy season 2019 showed a strong negative effect on the abundance of hog-nosed skunks (Figs 3 and 4 and S7 Table). Shrub cover (β = -1.36, BCI = -3.55 to 0.29) and coyote presence (β = -1.11, BCI = -2.71 to 0.04) had a negative effect but without strong support (their 95% BCIs were overlapped with zero) to explain the abundance of this species in the dry season 2020 (Fig 3 and S7 Table). On the other hand, the availability of small mammals had a strong positive effect on the abundance of pygmy skunks in the dry season 2019 (β = 0.63, BCI = 0.36 to 0.89), rainy season 2019 (β = 0.65, BCI = 0.48 to 0.83), and in the dry season 2020 (β = 0.56, BCI = 0.10 to 0.99). The abundance of this species was also positively related to the distance to the nearest water source in the dry season 2019 (β = 0.75, BCI = 0.27 to 1.29) but negatively to the shrub cover in the dry season 2020 (β = -1.65, BCI = -2.86 to -0.50), both relationships showed strong support (Figs 3 and 5 and S7 Table). Distance to the nearest water source (β = -0.24, BCI = -0.51 to 0.04) and presence of coyotes (β = -0.33, BCI = -0.72 to 0.00) had a negative effect but without strong support to explain the abundance of pygmy skunks in the rainy season 2019 (Fig 3 and S7 Table).

## Discussion

There is a lack of published information documenting the abundance and variation over space and time of American hog-nosed and pygmy spotted skunks throughout their range, as well as possible ecological factors that may affect them [27, 95]. In this regard, our research contributes to the knowledge of population ecology and highlights the relative importance of underlying factors that determine the abundance patterns of both skunks, which coexist sympatrically in a seasonal tropical forest at Huatulco National Park within the Mexican Pacific slope.

The R-N models may provide accurate estimates of abundance when their assumptions are met [43, 44, 67, 72, 73], so we took serious considerations in the sampling design and in the modeling framework (e.g. number of sites and sampling occasions). We also modeled detection probability and spatial variation in abundance as a function of biologically relevant covariates and accounted for spatial autocorrelation between sites for hog-nosed skunks, which are sources of bias that affect model performance and lead to unreliable estimates [42–44, 67].

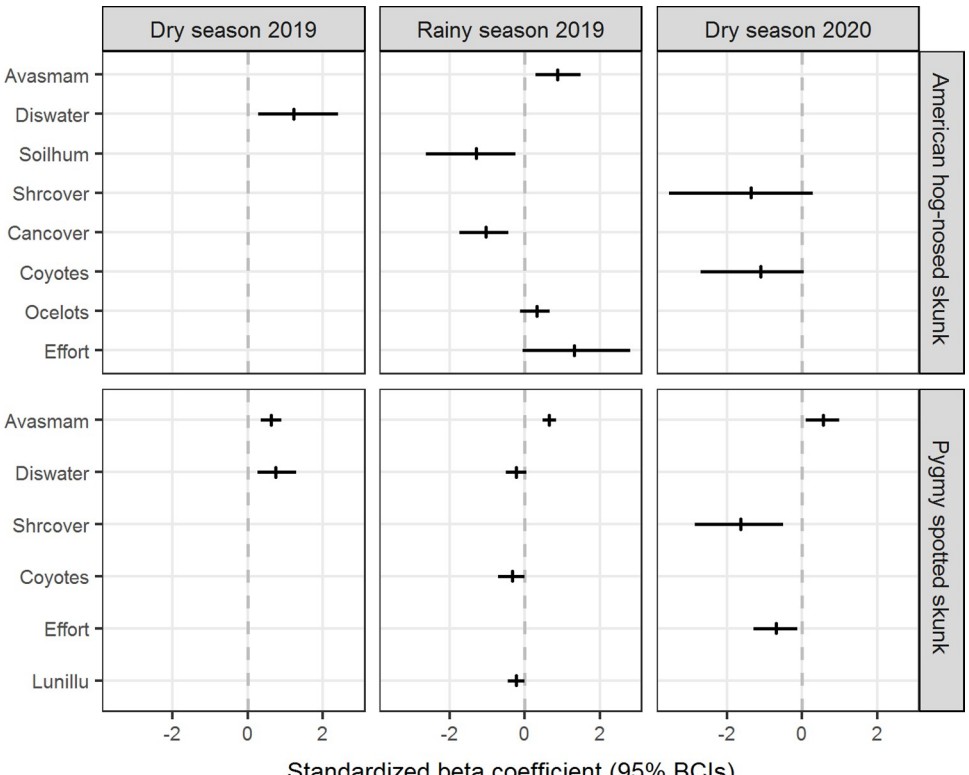

**Fig 3. Standardized beta coefficients showing the effect of the covariates from best-ranked R-N models on the abundance of skunk species during seasons surveyed.** The covariates should have a strong effect on abundance if the 95% Bayesian credible interval (BCI) of its coefficients did not overlap with zero. The vertical lines indicate the means, and the horizontal lines are 95% BCIs of the beta coefficients.

Although there was no evidence of spatial autocorrelation in the λ parameter estimate, we interpreted the average abundance per site of hog-nosed skunks as relative, assuming that individuals use four or five sites on average in our study zones. Nonetheless, we believe that the number of individuals is truly low, which resulted in few detections with several true zeros (i.e., sites where the species is absent [44, 76], at least in the protected zone) and, therefore, a lower detection probability (r = 0.02–0.09). The R-N models can estimate abundance with a positive bias between 10–22% due to detectability of less than 0.1 [43, 44, 71], but when fitted in a Bayesian framework have the potential to make more robust and reliable inferences in data sets with small sample sizes, few detections, and low detection probabilities [43, 67, 75, 81], for example species that occur in low densities [72, 74, 76]. These conditions in the R-N modeling allowed us to infer reasonably well the density of hog-nosed skunks (0.14–2.42 ind/km$^2$) at Huatulco National Park, within the same order of magnitude compared to previous estimations: 0.5–1.3 ind/km$^2$ in the Isthmus of Tehuantepec, Mexico [32, 40] and 2.6 ind/km$^2$ in west-central Texas, United States [55]. However, by considering the possible sources of bias, such as those mentioned, our study provides population information for the species with a better analytical approach.

Likewise, the R-N models typically produce unbiased estimates of the absolute abundance of target species when sampling is done at an appropriate spatial scale, such as the home range [71–74]. Our modeling results, thus, revealed for the first time the absolute abundance of pygmy skunks from detection-non-detection camera trap data using a statistically robust method. These findings showed ecologically realistic estimates with detection probabilities

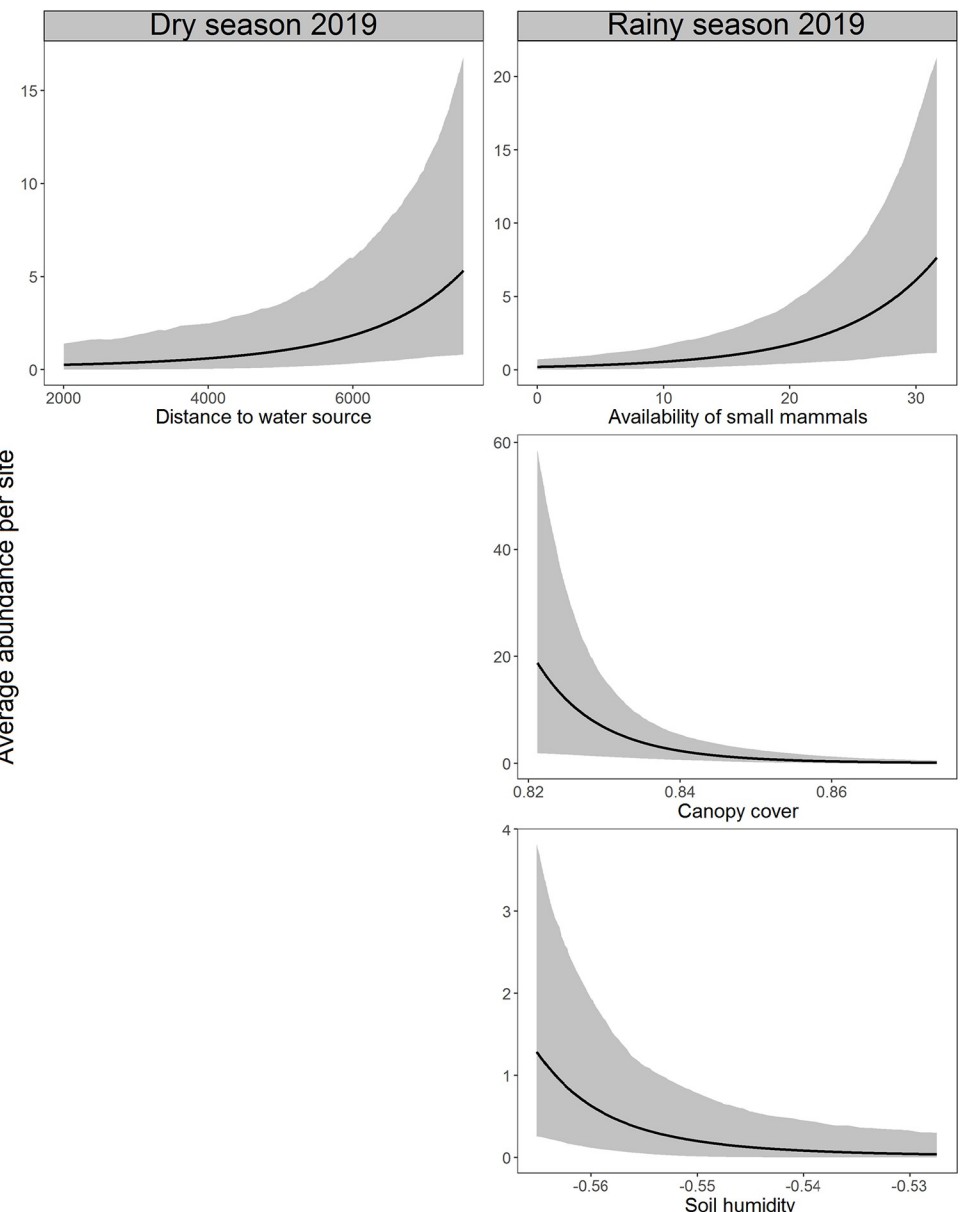

**Fig 4. Marginal effects plots of strongly supported covariates in the top-ranked R-N models on the abundance of hog-nosed skunks during seasons surveyed.** The focal covariate varies across its range of original values in each plot. The bold lines indicate posterior means, and the ribbons are 95% credible intervals.

greater than 0.1, allowing for density inferences in the study area comparable to those of other spotted skunk species. For instance, 9.0–19.0 ind/km$^2$ for the island skunks *S. gracilis amphiala* [20] and 5.02 ind/km$^2$ and 6.52–23.29 ind/km$^2$ for the eastern skunks *S. putorius* [96, 97] in regions of the United States. The correct use of the R-N model facilitated the estimation of the abundance of this elusive carnivore and showed evidence that it may be locally abundant in conserved or low-disturbance areas. Unfortunately, no further population studies are available on pygmy skunks, so data elsewhere in their range are required to compare the applicability of site-structured models for estimating unmarked populations.

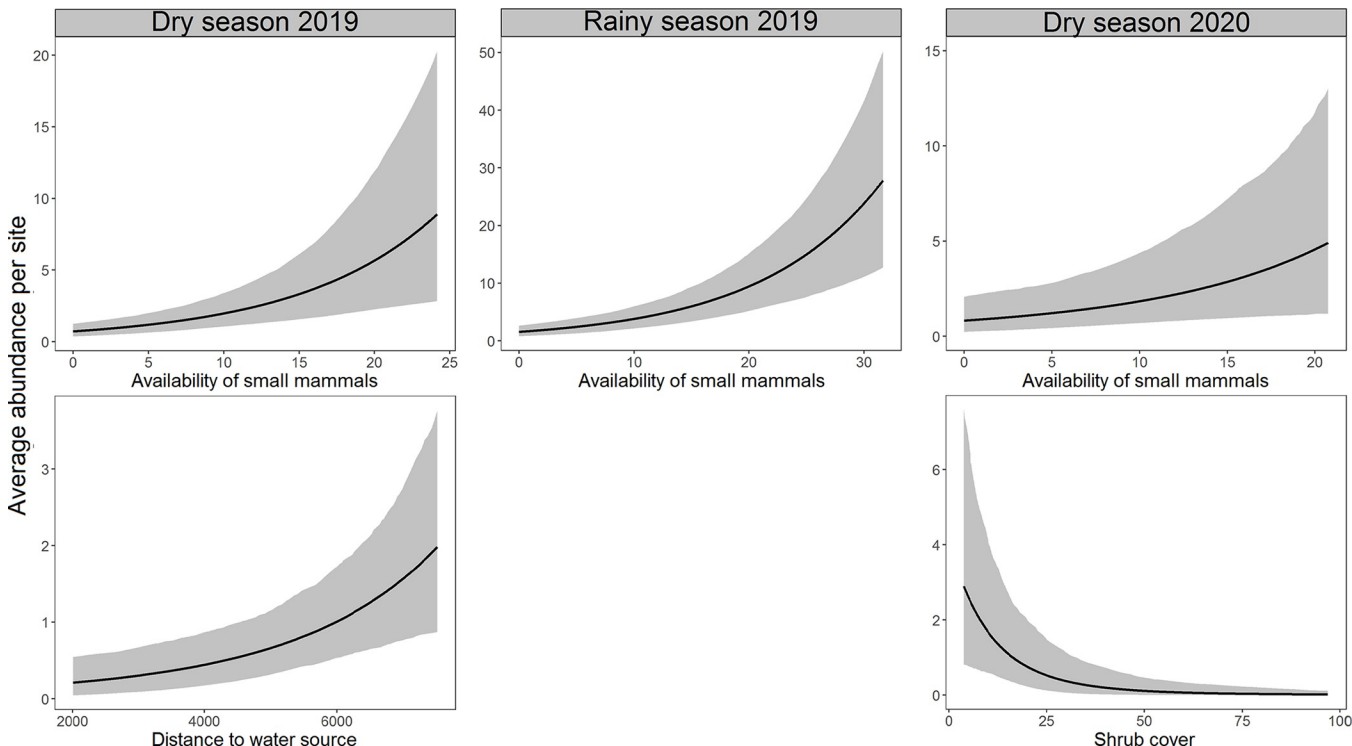

**Fig 5. Marginal effects plots of strongly supported covariates in the top-ranked R-N models on the abundance of pygmy skunks during seasons surveyed.**
The focal covariate varies across its range of original values in each plot. The bold lines indicate posterior means, and the ribbons are 95% credible intervals.

Competitive interactions influence abundance in multiple carnivore species dyads [3, 6, 11], with the body mass ratio of competitors being the primary trait determining their direction and strength [6, 7, 58]. It has been speculated that intraguild competition may explain changes in the abundances of sympatric skunks [24, 41, 95, 96], even recognizing an apparent negative relationship between species [24, 41] or with other small carnivores [14, 20]. We observed that the abundances of skunk species were positively related during the rainy season and negatively during the dry season at Huatulco National Park, as would be expected according to the predicted hypotheses. However, on the one hand, our data did not elucidate the driver on the positive effect among species abundances, and, otherwise, our regression models indicated that the negative effect of hog-nosed skunks on pygmy skunk populations was not informative. Moreover, the R-N models that incorporated the presence of the competing skunk as a covariate for pygmy skunk abundance showed low predictive accuracies. Both sets of results suggest that local abundance patterns of the small-sized species are not governed by the competitive dominance of the larger species, at least at our scale of analysis. Some research on mephitid assemblages in regions of North America infers that the largest members determine the intraguild dynamics, such as hog-nosed or striped (*M. mephitis*) skunks, having higher numbers [24, 41]. By contrast, pygmy skunks had higher abundance at the site level during the rainy season and were similar or slightly less abundant during the dry season than hog-nosed skunks in our study area. These findings possibly indicated some degree of ecological dominance of the subordinate species by being more abundant [9], as suggested for island spotted and hooded skunks in the United States [20, 25]. In any case, we did not reveal evidence of competitive pressure (i.e., suppression or exclusion at sampling sites) among skunk populations in line with previous studies [25, 32].

The strength of intraguild competition may be more intense at increasingly higher densities of interacting species [6, 10]. Hog-nosed skunk densities were low in the reserve studied, mainly in protected zone, and likely did not exceed the density threshold at which negative effects on pygmy skunk populations were observed, thus reflecting reduced competitive stress. Competition intensity in other carnivore guilds is weak when the superior competitor is absent or in low numbers, resulting in high densities of inferior competitors [13, 15, 18, 20], as could have occurred in our study system. Interestingly, a result that surprised us was the positive association among abundance per site of both skunks during the rainy season, to which we suggest that the observed effect was due to a behavioral rather than demographic response by pygmy skunks to hog-nosed skunk numbers. Large skunk species have more conspicuous aposematic coloration, and this conspicuousness increases the detection rate by potential predators [98, 99], so that the pygmy skunks may be more likely to go unnoticed, at least by larger carnivores, when they occur in sites with more hog-nosed skunks. This behavior is perhaps facilitated through fine-scale temporal partitioning by subordinate species to minimize agonistic encounters [41, 54]. Furthermore, antipredator benefits might enhance the intraguild competition effects, since skunk species may act as Müllerian mimics by sharing the same habitat [25].

Competitive interactions are also intensified when shared food resources are scarce [1, 3, 6], so that the superior species may suppress the inferior species by competing for prey [13, 100]. Although insect availability decreases considerably during the dry season in the tropical deciduous forest [34, 35], we observed that some secondary foods reported in the diet of the target species, such as small mammals, lizards, and fruits, were available during the drought in the environment studied. Research has shown a differentiation in prey size or percentage of each trophic category in pairs of sympatric skunk species and between mesocarnivores with a degree of insectivory [1, 25, 101]. The differential consumption of these food items, thus, probably reduced dietary competition among species for a limiting staple resource (i.e., insects) and allowed them to sustain their populations without evidence that hog-nosed skunks limited the numbers of pygmy skunks. Nevertheless, it has been found that the outcome of competitive interactions may be independent of resource dynamics [9], and the exploitation of shared prey is not a cause that triggers the intraguild competition effects [16, 58].

It is worth noting that the small, short-term spatiotemporal scale analyzed in our camera-trapping study might have limited the inferences about competitive interactions from the observed abundance patterns. For example, some studies have shown that intraguild competition effects on subordinate carnivore abundance can be modulated at the home range or ecoregion level regarding the dominant member´s range [16, 21], while others have recorded changes in abundance or population density among competing carnivores over long-term periods [13, 14, 18]. However, there is evidence that competition does not occur between sympatric skunk species, including closely related ones, at a local (southeastern Arizona) and regional (southwestern United States) spatial scale [25]. Locally, we did not document direct encounters between hog-nosed and pygmy skunks during our night fieldwork or in camera traps despite the sampling effort deployed, so we believe that the competitive relationships of these species can be shaped through scent markings. This idea is also supported by previous records in other mephitid assemblages indicating that interspecific interactions in nature are usually rare [24, 25], with occasional defense displays [27] and observations of individuals nearby without physical face-off [26, 32]. It is conceivable, therefore, that the rarity or inconsistency of interference prevents true competition from occurring or is negligible within the guild of insectivorous carnivores [5, 12].

The abundance of potential competitors could also reflect associations with food availability or differences in habitat preference [16, 22, 23], as suggested for sympatric skunks [32, 40, 41].

At Huatulco National Park, the small mammal availability was the most determinant driver of the skunk species abundance according to the best-ranked models. Small mammals are considered alternative prey for hog-nosed and pygmy skunks in periods of insect scarcity [29, 30], and the consistently high significance of their positive effects during the surveyed seasons suggested that this type of prey is abundantly or steadily available to both species. The spiny pocket mouse *Heteromys pictus* was the predominant rodent throughout the sampling period in our study area, with higher densities during the rainy season (unpublished data), which is in line with a species´ population study in areas surrounding the coast of Oaxaca, Mexico [102]. Yet, although these skunk species can food resource-switching, pygmy skunks have morphological adaptations that allow them to efficiently exploit small mammals (e.g., the pocket mouse) [27, 31, 103]. They are likely to have gained a competitive advantage by acquiring these prey items, similar to other small carnivores [20, 21], and play the role of a superior exploitative competitor over the shared resource [9], which led to this factor better explaining their abundance. In this regard, our findings support that the food availability drives spatio-temporal variations in skunk species abundance within a temporally variable source-sink dynamic [25, 40, 41], and if so, it may decrease the potential for competitive interactions [104, 105].

Other bottom-up predictors also contributed to explaining the abundance patterns of concerned species, including robust associations either positive with distance to the nearest water source or negative with canopy and shrub cover. The low abundance of both skunks at sites closest to water bodies in our study area may be due to a higher likelihood of negative interactions with potential predators, mainly during water scarcity periods [106, 107]. Intraguild predation particularly has a direct impact on populations of small carnivores [5, 8, 11], so the abundance of hog-nosed and pygmy skunks may represent a trade-off between the need to satisfy their requirements for water and avoid lethal encounters with larger carnivores. These species may have been able to obtain water from their food and small temporary natural reservoirs, as occurs in other seasonally dry and arid regions [27, 29, 95, 108], and visit water bodies less frequently, which could result in fewer skunks at those sites.

The most predictively accurate R-N models further indicated that hog-nosed and pygmy skunks were more abundant in areas with less vegetation cover, both arboreal during the rainy season and shrubby during the dry season, respectively. Both relationships could largely be explained because skunk species have antipredator defense mechanisms [27, 98, 99], which more successfully deter terrestrial predators such as coyotes in open areas where skunks are more susceptible to ambush attacks [109, 110]. While avian predators such as owls are another possible source of mortality for skunks [14, 110, 111], the species we studied are more active on cloudier nights [54] and, additionally, pygmy skunks showed increased detectability when there was lower lunar illumination during the rainy season. These nocturnal conditions presumably offer them some protection from raptors. Therefore, it is likely that skunk predation by terrestrial and aerial predators is not frequent enough and that there are higher numbers of individuals in areas with sparse vegetative cover, depending on seasonality.

More broadly, we found that local abundance patterns of hog-nosed and pygmy skunks were determined primarily by the availability of alternative prey rather than intraguild competition, suggesting that bottom-up predictors were significant for promoting coexistence among both species at Huatulco National Park. Our findings further fitted the prevailing pattern of local-scale coexistence recorded in other mephitid assemblages, in which sympatric skunks coexist by presenting spatiotemporal variations in their relative abundances or population densities in tropical habitats of Mexico [32, 40, 41]. Nonetheless, we highlight that explicit consideration of the scale at which target skunk species move is advisable when investigating intraguild interactions. The development of dynamic hierarchical models of interacting species

at different spatial scales would therefore enhance inferences on abundance patterns within this carnivore guild with more informative parameters from camera trap data (e.g., [112]).

## Conservation implications

Populations of the hog-nosed and pygmy skunks are currently experiencing a decline, attributed primarily to habitat loss and interspecific interactions [38, 39]. Our data corroborates this general population trend for hog-nosed skunks while showing high density for endemic pygmy skunks at Huatulco National Park. To support the results, we also provide insight into the underlying factors that determine the local abundance of these understudied and threatened carnivores. This knowledge will improve our understanding of the conditions or requirements necessary to maintain and recover populations of both skunks, as well as the mechanisms that govern their coexistence in a seasonal environment. So, it would be pertinent for the management and conservation program of this protected natural area to consider the interplay of the most important factors, including human-induced changes (e.g., the presence of feral dogs) due to their potential short-term cascading effects, to direct conservation efforts for concerned species effectively. Further studies are needed to assess how intraguild interactions, resource availability, and habitat complexity influence abundance patterns of skunk species in other regions where they are sympatry based on ecologically appropriate spatial and temporal sampling scales.

## Supporting information

**S1 Dataset. Database of independent detections of hog-nosed and pygmy skunks in Huatulco National Park, Oaxaca, Mexico.**
(CSV)

**S1 Table. Variance inflation factor (VIF) values for covariates used in the modeling framework at the three surveyed seasons.**
(DOCX)

**S2 Table.** Selection of candidate Bayesian Generalized Linear Models explaining the effect of surveyed season and study zone on the abundances of skunk species (A) and the relationship in abundances between skunk species during the surveyed seasons in each study zone (B) using leave-one-out cross-validation for pairwise model comparisons.
(DOCX)

**S3 Table. Parameter estimates of the best-ranked Bayesian Generalized Linear Models explaining the relationships in abundance between skunk species during the surveyed seasons in each study zone.**
(DOCX)

**S4 Table. Selection of candidate Royle-Nichols models explaining the detection probability (r) for skunk species during the surveyed seasons using leave-one-out cross-validation for pairwise model comparisons.**
(DOCX)

**S5 Table. Selection of candidate Royle-Nichols models explaining the abundance ($\lambda$) of skunk species from a priori hypotheses by three subsets of variables during the surveyed seasons using leave-one-out cross-validation for pairwise model comparisons.**
(DOCX)

**S6 Table. Parameter estimates of spatial models incorporating the spatial random effect to determine the average abundance per site of hog-nosed skunks at the each surveyed season.** (DOCX)

**S7 Table. Parameter estimates of top-ranked Royle-Nichols models explaining the abundance (λ) of skunk species at the each surveyed season.** (DOCX)

## Acknowledgments

We are grateful to the Comisión Nacional de Áreas Naturales Protegidas (CONANP), especially to the authorities of the Huatulco National Park for the permits and facilities to carry out this study, as well as the park rangers for their logistical support. We thank the Animal Ecology Laboratory colleagues for their assistance and help during the fieldwork and G. Pérez-Irineo and D. Mondragón for reviewing different versions of the document. We would also like to thank the Secretaría de Medio Ambiente y Recursos Naturales (SEMARNAT) for providing the scientific collection licenses for teaching purposes in the field of wildlife (SGPA/DGVS/008795/18 and SGPA/DGSV/11153/19).

## Author Contributions

**Conceptualization:** Alejandro Hernández-Sánchez, Antonio Santos-Moreno.

**Data curation:** Alejandro Hernández-Sánchez.

**Formal analysis:** Alejandro Hernández-Sánchez, Antonio Santos-Moreno.

**Funding acquisition:** Antonio Santos-Moreno.

**Investigation:** Alejandro Hernández-Sánchez.

**Methodology:** Alejandro Hernández-Sánchez, Antonio Santos-Moreno.

**Project administration:** Alejandro Hernández-Sánchez.

**Supervision:** Antonio Santos-Moreno.

**Validation:** Antonio Santos-Moreno.

**Visualization:** Alejandro Hernández-Sánchez.

**Writing – original draft:** Alejandro Hernández-Sánchez, Antonio Santos-Moreno.

**Writing – review & editing:** Antonio Santos-Moreno.

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
