## [Decision Letter · Decision Letter 0]

22 Nov 2023

PONE-D-23-33252Availability of alternative prey rather than intraguild interactions determines the local abundance of two understudied and threatened small carnivore speciesPLOS ONE

Dear Dr. Santos-Moreno,

Thank you for submitting your manuscript to PLOS ONE. After careful consideration, we feel that it has merit but does not fully meet PLOS ONE’s publication criteria as it currently stands. Therefore, we invite you to submit a revised version of the manuscript that addresses the points raised during the review process.

We look forward to receiving your revised manuscript.

Kind regards,

Luca Nelli, PhD

Academic Editor

PLOS ONE

Journal Requirements:

   "Instituto Politécnico Nacional de México provide economic support and a graduate student  grant to the first author."

5. We note that Figure 1 in your submission contain map/satellite images which may be copyrighted. All PLOS content is published under the Creative Commons Attribution License (CC BY 4.0), which means that the manuscript, images, and Supporting Information files will be freely available online, and any third party is permitted to access, download, copy, distribute, and use these materials in any way, even commercially, with proper attribution. For these reasons, we cannot publish previously copyrighted maps or satellite images created using proprietary data, such as Google software (Google Maps, Street View, and Earth). For more information, see our copyright guidelines: http://journals.plos.org/plosone/s/licenses-and-copyright.

Additional Editor Comments:

Please make sure to address the reviewers comments, with respect in particular to main caveat in the analysis: the low number of detections of the hog-nosed skunk. Additionally, please make sure that you address reviewer's 2 comment on expansion of literature review.

Reviewers' comments:

Reviewer's Responses to Questions

**Comments to the Author**

1. Is the manuscript technically sound, and do the data support the conclusions?

Reviewer #1: Yes

Reviewer #2: Yes

2. Has the statistical analysis been performed appropriately and rigorously? 

Reviewer #1: No

Reviewer #2: Yes

3. Have the authors made all data underlying the findings in their manuscript fully available?

Reviewer #1: No

Reviewer #2: No

4. Is the manuscript presented in an intelligible fashion and written in standard English?

Reviewer #1: Yes

Reviewer #2: Yes

5. Review Comments to the Author

Reviewer #1: This article provides a clear look at abundances and interactions of two declining. skunk species. They do a good job of applying statistical techniques and placing their results in the context of other literature. I have a few minor recommendations related to their statistics. I think this article is adding valuable information about relatively understudied mammal species.

Line 71-73: This sentence surprised me because I would expect jaguars to be the largest neotropical felid. I see that you’re specifying “small neotropical felids” though. Maybe specify as “small neotropical felids (<## kg)? Or whatever the distinguishing feature is.

Methods

Line 181: Could probably just say (Table 1) for the reference. It wasn’t clear to me on the first read through that all covariate information was there and not just calculations

Table 1: The range column confused me for the presence/absence in particular. Maybe 0-2 and 0-58 detections? Or observations? Records sounds like rows in a data frame and I don’t think that’s what you’re getting at. In addition, I think that the mean or median and standard error would be a good addition to this column, as that’s often more useful than the range. The reader can’t tell if the values in the range are unusually high/low compared to the rest of the data.

Table 1: I like that you include your predictions in the table

Line 260-261: Goodness of fit is often conducted on the global model (model with all covariates) rather than the top-ranked model. Take a look at your global model to make sure it also looks ok

Line 270: Outlier removal is challenging. If these outliers were not representative of the rest of your data, then removing them makes sense. For example, if skunks were released from a rehab facility into an area, it might have an unusually high abundance that might not be representative of the area. But if the outliers are representative – an area just has a lot of skunks compared to other areas – then you often want to retain them. I’m using skunk counts as an example but the same concept for any variable.

Line 271: It seems odd to me that you use the R-N model and then use non-parametric tests. It’s not wrong, but you’re probably sacrificing a decent amount of statistical power (ability to estimate abundance and find differences). Ideally, you would be able to test differences among study areas and seasons based on R-N output and post-hoc comparisons. You’d do that by having all the data in one model, with season, study area, and year as covariates. I recommend checking to see if that’s possible. If not, I would think you could use parametric tests rather than non-parametric. Models are relatively robust to the assumption of paired vs. unpaired data, so I would not recommend sacrificing statistical power to use the non-parametric test here. That said, this approach isn’t wrong, just not ideal.

Line 275: I would make it clearer that the relative abundances are for hog-nosed skunks while absolute is for pygmy skunks. On Line 218, you mention that you interpret abundances as relative abundances rather than absolute abundances due to the potential of non-independent observations. Consider stating here that you can calculate absolute abundances for pygmy skunks. In line 275, you calculate absolute abundances for pygmy skunks. Consider mentioning that you only have relative abundances for hog-nosed skunks. Also, it might be good to provide the effective area of sampling for pygmy skunks here, or how you calculated it, for reproducibility.

Line 279: I’m surprised at the choice of gamma distribution. Population abundance is generally discrete, and therefore modeled with a Poisson or negative binomial distribution. Are you using the population abundances output from your R-N model (which means they might be continuous)? If so, please make that clearer.

Results

Line 297: Consider stating your abbreviations H and W here in the results instead of in the methods. It’s hard to remember that between sections when it’s not a common abbreviation.

Line 300: Please double-check the W value – in parametric tests, a difference of 0 would mean no difference, versus the p<0.001 described here. The previous one with a p<0.001 had a W=421. It might be fine, but please double-check.

Line 303: Consider adding the relative abundances of hog-nosed snakes alongside the pygmy skunk density estimates.

Line 312: These results are a bit confusing given the difference between the interpretation and the coefficients. The ones that are written as negative relationships have positive coefficients (e.g. 0.04), while the ones with positive relationships have negative coefficients (e.g. -4.93). This seems unusual, and might be worth a sentence in the methods explaining why the sign of the coefficient (+/-) and the interpretation don’t match. I don’t know why they’d be reversed like this, so please make sure everything is ok here and that there’s a reason for the signs.

Line 313: Your credible interval abbreviation is BCI rather than BIC. Please make sure to fix throughout. BIC is also an abbreviation for Bayesian Information Criteria.

Line 317 and Figure 3: I think you should use caution in interpreting the rainy season protected zone data. The relationship has a decent effect size (coefficient), but the figure indicates that the trend is driven by the one point with high abundances of both skunk species. Consider adding a sentence mentioning that caution.

Line 327: I think you mean rainy season instead of wet season here, for consistency.

Figure 4: Please add measure of uncertainty (95% Bayesian Credible Interval) to the x axis label

Figure 5: I’m confused by the multiple “availability of small mammals” graphs for the pygmy spotted skunk. I think it would be helpful to specify the season in the figure – you can have multiple variables in your facets, perhaps something like facet_wrap(~ species + season).

Supplementary Info

I encourage you to include table captions and key to abbreviations for each table in the supplementary information documents. It’d make them easier to understand.

S1 and S2: Please write captions for these tables

S3 Table: I don’t see any null models in this table. I think it would make sense to include one for this stage

Reviewer #2: General comments:

Dear Authors,

This is a well-written and well-referenced manuscript, and an very interesting read. The work in it presents an analysis from camera trapping grid in a tropical region in Mexico. The design is optimised for pygmy skunks, but an attempt is made to obtain relative indices for the larger American hog-nosed skunk to infer competitive relationships between the two, as well as the importance of predation by felids and coyotes, and the relative strength of such interactions compared to food availability, and habitat structure.

The manuscript is well structured, and the methods well described, though I have highlighted a few points, mostly for clarity’s sake, below. The main caveat here is the small number of detections of hog-nosed detections in general, but particularly in the protection zone, which in combination with model violations of independent detections between stations for this same species limits inference. This is noted in the manuscript, but it is unclear if it is taken into consideration when comparing with previous literature or making inferences about the interactions among the two target species. How to address multi-scale processes is a challenge in empirical ecology. Kleiven et al. (2023) attempt to address this, but I understand it may be outside the scope of this MS, and it would not address the low detections. Nonetheless, hog-nosed skunks may be a suitable covariate in the pygmy skunk analysis regardless of the above, which could be noted.

More generally, this work is refreshing, as it highlights that the occurrence of intraguild interactions that lead to some form of suppression among guild members may not be as common relative to the diversity of species and community assemblages and their seasonal and multi-annual dynamics as the recent literature on the topic seems to suggest. Along this lines, and since this work is set within this wider set of the literature linked to intraguild interactions, which includes competition but not only, additional discussion of this results against such literature would be welcome. See comment L446 below.

Alternatively, or additionally, this MS would benefit from discussing the rich literature of competitive interactions, particularly since there is evidence that these species may not be interacting through any other means. Discussion of references such as Rosenzweig (1966), MacArthur and Levins (1967), Aunapuu et al. (2010), Monterroso et al. (2020), and references within would be beneficial.

I have compiled a list of more specific comments below.

References

Aunapuu, M., Oksanen, L., Oksanen, T., and Korpimaki, E. (2010). Intraguild predation and interspecific co-existence between predatory endotherms. Evol Ecol Res 12: 151-168

Kleiven, E.F., Barraquand, F., Gimenez, O. et al. A Dynamic Occupancy Model for Interacting Species with Two Spatial Scales. JABES 28, 466–482 (2023). https://doi.org/10.1007/s13253-023-00533-6

MacArthur, R. and Levins, R (1967). The limiting Similarity, Convergence, and Divergence of Coexisting Species. The Americna Naturalist, 101(921) https://doi.org/10.1086/282505

Monterroso, P., Díaz-Ruiz, F., Lukacs, P. M., Alves, P. C., and Ferreras, P.. 2020. Ecological traits and the spatial structure of competitive coexistence among carnivores. Ecology 101(8):e03059. 10.1002/ecy.3059

Rosenzweig, M. L., Community Structure in Sympatric Carnivora, Journal of Mammalogy, Volume 47, Issue 4, 2 December 1966, Pages 602–612, https://doi.org/10.2307/1377891

Specific Comments:

Abstract:

L23: ‘inverse’ is correct but perhaps unnecessarily mathematical for a lay audience which is more likely to read the abstract, ‘negative’ would be equally suitable.

L40-41: wording after the comma ‘with the highest…’ could be made clearer that it refers to the prey availability and habitat complexity models.

Introduction:

L57: killing is not necessarily followed by consumption, ‘intraguild killing’ may be more appropriate.

Methods:

L174: What is considered an independent detection? i.e., how long between consecutive visits before it’s considered independent?

L203: Table 1 is a great idea. For lunar illumination, how is this information produced? Is it a linear function of the moon phase? Does it take into consideration cloud or vegetation cover?

L210: Is there a reason for selecting 120 nights over other lengths?

L213: this sentence could be clearer that it refers to the three surveys (per site?) of 120 trap nights each. Also, table 2 referenced here seems to relate to the full effort, rather than the described data used.

L221: It would be valuable to have a statement here about the method’s ability to perform with few detections, as both seasons in the protection zone only had 3 detections of hog-nosed skunks. I see this is partly addressed in line 238, but the question of whether the model performs well at such low number of detections remains valid.

L223: Because the table seems to show the full surveying effort, rather than the effort used for analysis, it may be misleading. In 2020 disturbed zone the number of detections seems to half for both species, but this is likely explained by the lower effort.

L244: No thinning necessary?

L270: This point should be elaborated. Is it sensible to remove datapoints? How is an outlier defined? What influence does it have on the results?

L279: It is unclear what (scale) the regression is applied to. As per the above section there are a max of 6 abundance estimates per species. Are the regressions fit to these 6 estimates? This is clearly not the case given that covariates are gathered at station level and the figure outputs, but it is not clear from this paragraph.

L282: does ‘abundance’ (discrete) refer to density (continuous)? If not, is a gamma distribution preferred over a poisson?

L283: ‘Similar to R-N modelling’?

Results:

L312-318: Unclear what beta’s stand for, if slope, in the transformed or back transformed scale? Please clarify as otherwise I would expect the signs of the parameters to match those of the text (positive and negative).

Discussion:

L413-420: What are the implications of the low number of detections and model violations for hog-nosed skunks? Particularly when comparing with other studies. Is this study more or less likely than the cited literature to offer robust estimates?

L446: Since this research has been set in the context of intraguild interactions, namely competition but also killing and fear, it would suit to comment on the fact that interactions that may lead to any form of suppression of the subordinate species are not as common as it may transpire from high-impact publications. Alternatively, see general comment on paper focus.

6. PLOS authors have the option to publish the peer review history of their article (what does this mean?). If published, this will include your full peer review and any attached files.

Reviewer #1: No

Reviewer #2: No

---

## [Author Response · Author response to Decision Letter 0]

17 Jul 2024

Author's Response to Reviewer

Observation/Comment Response/Reply

Abstract 

L23: ‘inverse’ is correct but perhaps unnecessarily mathematical for a lay audience which is more likely to read the abstract, ‘negative’ would be equally suitable. Done without problem.

(Line:23)

L40-41: wording after the comma ‘with the highest…’ could be made clearer that it refers to the prey availability and habitat complexity models. Done without problem. 

(Line:40-41)

Introduction 

L57: killing is not necessarily followed by consumption, ‘intraguild killing’ may be more appropriate. Done without problem. Interspecific killing may not result in consumption of the victim species, but it is considered an extreme form of interference competition, so we considered it appropriate.

(Line: 58)

Line 71-73: This sentence surprised me because I would expect jaguars to be the largest neotropical felid. I see that you’re specifying “small neotropical felids” though. Maybe specify as “small neotropical felids (<## kg)? Or whatever the distinguishing feature is. Done without problem. 

(Line: 72-73)

Methods 

We note that Figure 1 in your submission contain map/satellite images which may be copyrighted. All PLOS content is published under the Creative Commons Attribution License (CC BY 4.0), which means that the manuscript, images, and Supporting Information files will be freely available online, and any third party is permitted to access, download, copy, distribute, and use these materials in any way, even commercially, with proper attribution. Done without problem. The vector datasets in Fig 1 were freely downloaded online in shapefile format from official websites.

(Line:143-150, 155-156)

L174: What is considered an independent detection? i.e., how long between consecutive visits before it’s considered independent? Done without problem. 

(Line:183-185)

Line 181: Could probably just say (Table 1) for the reference. It wasn’t clear to me on the first read through that all covariate information was there and not just calculations Done without problem.

(Line: 192)

Table 1: The range column confused me for the presence/absence in particular. Maybe 0-2 and 0-58 detections? Or observations? Records sounds like rows in a data frame and I don’t think that’s what you’re getting at. In addition, I think that the mean or median and standard error would be a good addition to this column, as that’s often more useful than the range. The reader can’t tell if the values in the range are unusually high/low compared to the rest of the data.

Table 1: I like that you include your predictions in the table Done without problem. We included the mean and standard deviation values.

(Line:199)

Table 1 is a great idea. For lunar illumination, how is this information produced? Is it a linear function of the moon phase? Does it take into consideration cloud or vegetation cover? Done without problem. We included this information in the lunar illumination row.

(Line:199)

L210: Is there a reason for selecting 120 nights over other lengths? Done without problem. 

(Line:224-226)

L213: this sentence could be clearer that it refers to the three surveys (per site?) of 120 trap nights each. Also, table 2 referenced here seems to relate to the full effort, rather than the described data used. Done without problem. We clarify this sentence. Table 2 refers to the data used and we attached the data set as a complementary file. (Line:222-224, 233)

L218: I would make it clearer that the relative abundances are for hog-nosed skunks while absolute is for pygmy skunks. On Line 218, you mention that you interpret abundances as relative abundances rather than absolute abundances due to the potential of non-independent observations. Consider stating here that you can calculate absolute abundances for pygmy skunks. In line 275, you calculate absolute abundances for pygmy skunks. Consider mentioning that you only have relative abundances for hog-nosed skunks. Done without problem. For a better understanding, we added additional information.

(line:238-245)

L221: It would be valuable to have a statement here about the method’s ability to perform with few detections, as both seasons in the protection zone only had 3 detections of hog-nosed skunks. I see this is partly addressed in line 238, but the question of whether the model performs well at such low number of detections remains valid. Done without problem.

(Line:248-257)

L223: Because the table seems to show the full surveying effort, rather than the effort used for analysis, it may be misleading. In 2020 disturbed zone the number of detections seems to half for both species, but this is likely explained by the lower effort. The table shows the sampling effort used in the analysis, therefore effort was a covariate for the probability of detection. For a better understanding, we added additional information.

(Line:227-231)

L244: No thinning necessary? There's no need

Line 260-261: Goodness of fit is often conducted on the global model (model with all covariates) rather than the top-ranked model. Take a look at your global model to make sure it also looks ok Done without problem. We checked and made sure the global model also had a good fit.

L270: This point should be elaborated. Is it sensible to remove datapoints? How is an outlier defined? What influence does it have on the results? Done without problem. We assessed this point with robust data analysis.

(Line:328-335)

Line 270: Outlier removal is challenging. If these outliers were not representative of the rest of your data, then removing them makes sense. For example, if skunks were released from a rehab facility into an area, it might have an unusually high abundance that might not be representative of the area. But if the outliers are representative – an area just has a lot of skunks compared to other areas – then you often want to retain them. I’m using skunk counts as an example but the same concept for any variable. Done without problem. We assessed this point with robust data analysis.

(Line:328-335)

Line 271: It seems odd to me that you use the R-N model and then use non-parametric tests. It’s not wrong, but you’re probably sacrificing a decent amount of statistical power (ability to estimate abundance and find differences). Ideally, you would be able to test differences among study areas and seasons based on R-N output and post-hoc comparisons. You’d do that by having all the data in one model, with season, study area, and year as covariates. I recommend checking to see if that’s possible. If not, I would think you could use parametric tests rather than non-parametric. Models are relatively robust to the assumption of paired vs. unpaired data, so I would not recommend sacrificing statistical power to use the non-parametric test here. That said, this approach isn’t wrong, just not ideal. Done without problem. We followed your recommendation and used Generalized Linear Models (GLMs) in a Bayesian framework. (Line: 313-316)

Line 275: It might be good to provide the effective area of sampling for pygmy skunks here, or how you calculated it, for reproducibility. Done without problem. For a better understanding, we added additional information for reproducibility.

(Line:339-348)

Line 279: I’m surprised at the choice of gamma distribution. Population abundance is generally discrete, and therefore modeled with a Poisson or negative binomial distribution. Are you using the population abundances output from your R-N model (which means they might be continuous)? If so, please make that clearer. Done without problem. We clarify this information and used GLMs in a Bayesian framework with Poisson and Negative Binomial error distribution.

(Line:313-317)

L279: It is unclear what (scale) the regression is applied to. As per the above section there are a max of 6 abundance estimates per species. Are the regressions fit to these 6 estimates? This is clearly not the case given that covariates are gathered at station level and the figure outputs, but it is not clear from this paragraph. Done without problem.

(Line:311-312)

L282: does ‘abundance’ (discrete) refer to density (continuous)? If not, is a gamma distribution preferred over a poisson? Done without problem. We clarify this information and used GLMs in a Bayesian framework with Poisson and Negative Binomial error distribution.

(Line:313-317)

L283: ‘Similar to R-N modelling’? Done without problem

(Line:320-324)

Results 

Line 297: Consider stating your abbreviations H and W here in the results instead of in the methods. It’s hard to remember that between sections when it’s not a common abbreviation. Because we used other statistical methods, these abbreviations no longer appear in the results section.

(Line:352-362)

Line 300: Please double-check the W value – in parametric tests, a difference of 0 would mean no difference, versus the p<0.001 described here. The previous one with a p<0.001 had a W=421. It might be fine, but please double-check. We used other statistical methods.

(Line:352-362)

Line 303: Consider adding the relative abundances of hog-nosed snakes alongside the pygmy skunk density estimates. Done without problem. 

(Line:382-386)

Line 312: These results are a bit confusing given the difference between the interpretation and the coefficients. The ones that are written as negative relationships have positive coefficients (e.g. 0.04), while the ones with positive relationships have negative coefficients (e.g. -4.93). This seems unusual, and might be worth a sentence in the methods explaining why the sign of the coefficient (+/-) and the interpretation don’t match. I don’t know why they’d be reversed like this, so please make sure everything is ok here and that there’s a reason for the signs. Done without problem. We checked the interpretation and coefficients of the relationships for a better understanding.

(Line:374-380)

Line 313: Your credible interval abbreviation is BCI rather than BIC. Please make sure to fix throughout. BIC is also an abbreviation for Bayesian Information Criteria. It was done without problem in all the Results section.

(Line: 351-386)

Line 317 and Figure 3: I think you should use caution in interpreting the rainy season protected zone data. The relationship has a decent effect size (coefficient), but the figure indicates that the trend is driven by the one point with high abundances of both skunk species. Consider adding a sentence mentioning that caution. We considered your suggestion for interpreting this result in the text and included the 95% BCIs in Figure 2 for better support. (Line:380-382)

L312-318: Unclear what beta’s stand for, if slope, in the transformed or back transformed scale? Please clarify as otherwise I would expect the signs of the parameters to match those of the text (positive and negative). Done without problem. We checked the interpretation and coefficients of the relationships for a better understanding.

(line:374-380)

Line 327: I think you mean rainy season instead of wet season here, for consistency. Done without problem 

(Line:396)

Figure 4: Please add measure of uncertainty (95% Bayesian Credible Interval) to the x axis label Done without problem in the Figure 3

Figure 5: I’m confused by the multiple “availability of small mammals” graphs for the pygmy spotted skunk. I think it would be helpful to specify the season in the figure – you can have multiple variables in your facets, perhaps something like facet_wrap(~ species + season). Because the number of plots for each season and species varies, we decided to make a separate figure for each species specifying the season for greater clarity (Figs 4 and 5).

(Line: 473-481)

Discussion 

L413-420: What are the implications of the low number of detections and model violations for hog-nosed skunks? Particularly when comparing with other studies. Is this study more or less likely than the cited literature to offer robust estimates? We addressed this point in more detail. (Line:490-510)

L446: Since this research has been set in the context of intraguild interactions, namely competition but also killing and fear, it would suit to comment on the fact that interactions that may lead to any form of suppression of the subordinate species are not as common as it may transpire from high-impact publications. Alternatively, see general comment on paper focus. We deepen our discussion on intraguild interactions in the context of interspecific competition and the suppression of subordinate species.

(Line: 524-590)

The manuscript is well structured, and the methods well described, though I have highlighted a few points, mostly for clarity’s sake, below. The main caveat here is the small number of detections of hog-nosed detections in general, but particularly in the protection zone, which in combination with model violations of independent detections between stations for this same species limits inference. This is noted in the manuscript, but it is unclear if it is taken into consideration when comparing with previous literature or making inferences about the interactions among the two target species We included information on the method's ability to perform with few detections supported by published literature. We also analyzed spatial autocorrelation to account for the violation of the assumption of independence of detections between stations in the hog-nosed skunk using a statistically and computationally efficient method (Restricted Spatial Regression method extended to single-season occupancy models). We rewrote the text in the Discussion section to better compare our results with previous literature (Line: 490-523).

Supplementary Info 

I encourage you to include table captions and key to abbreviations for each table in the supplementary information documents. It’d make them easier to understand. Done without problem.

(Line: 965-992)

S1 and S2: Please write captions for these tables Done without problem in all tables.

S3 Table: I don’t see any null models in this table. I think it would make sense to include one for this stage We understood the point but included the null model in Table 4 where we compared the best models of each subset of models.

Other changes 

We rewrote part of the text throughout the manuscript, mainly the Results (Abundance of skunks) and Discussion sections.

We included the suggested literature and cited other articles in the Discussion section to support the results.

We edited the figures to keep the same format.

We included more complementary information for greater support and clarity.

---

## [Decision Letter · Decision Letter 1]

7 Aug 2024

PONE-D-23-33252R1Availability of alternative prey rather than intraguild interactions determines the local abundance of two understudied and threatened small carnivore speciesPLOS ONE

Dear Dr. Santos-Moreno,

Thank you for submitting your manuscript to PLOS ONE. After careful consideration, we feel that it has merit but does not fully meet PLOS ONE’s publication criteria as it currently stands. Therefore, we invite you to submit a revised version of the manuscript that addresses the points raised during the review process. Both reviewers were pleased with the edits, and I agree that the manuscript has now much improved. Please make sure to address the final minor comments from reviewer #1, and I'll be happy to proceed with acceptance.

We look forward to receiving your revised manuscript.

Kind regards,

Luca Nelli, PhD

Academic Editor

PLOS ONE

Journal Requirements:

Reviewers' comments:

Reviewer's Responses to Questions

**Comments to the Author**

1. If the authors have adequately addressed your comments raised in a previous round of review and you feel that this manuscript is now acceptable for publication, you may indicate that here to bypass the “Comments to the Author” section, enter your conflict of interest statement in the “Confidential to Editor” section, and submit your "Accept" recommendation.

Reviewer #1: (No Response)

Reviewer #2: All comments have been addressed

2. Is the manuscript technically sound, and do the data support the conclusions?

Reviewer #1: Yes

Reviewer #2: Yes

3. Has the statistical analysis been performed appropriately and rigorously? 

Reviewer #1: Yes

Reviewer #2: Yes

4. Have the authors made all data underlying the findings in their manuscript fully available?

Reviewer #1: Yes

Reviewer #2: Yes

5. Is the manuscript presented in an intelligible fashion and written in standard English?

Reviewer #1: Yes

Reviewer #2: Yes

6. Review Comments to the Author

Reviewer #1: Great job addressing comments from the past review. I have a couple of small comments regarding the explanations of the negative binomial and Poisson distributions, which are hopefully easy to address.

Lines 313-314: Did you only use the negative binomial, or were there some that you used the Poisson for? Or did you decide based on AIC/other criteria? It’s not quite clear to me. Only NB is mentioned in the following paragraph, but please mention the Poisson if you fit some models with that distribution, or how you chose between the two for each model.

Line 317-319: Close on the explanation for NB instead of Poisson – note that the Poisson can’t explain extra variability, so that’s why we consider other distributions for a lot of highly variable ecological data. Consider this wording instead: “The Negative Binomial distribution includes a dispersion parameter that allows it to explain more variability than the Poisson distribution”

Results: Minor comment, but consider having BCI’s as -0.5 to -0.2 instead of -0.5 - -0.2. The style of the dashes makes it hard to pick out the negatives in the second number. Alternatively or in addition, you could add a + before positive numbers. But also ok if you choose to leave this as-is.

Results: Great job improving your results section.

Reviewer #2: The comments have been addressed thoroughly and I am happy to recommend its acceptance for publication.

7. PLOS authors have the option to publish the peer review history of their article (what does this mean?). If published, this will include your full peer review and any attached files.

Reviewer #1: No

Reviewer #2: No

---

## [Author Response · Author response to Decision Letter 1]

21 Aug 2024

Methods:

We clarified this point and added information for a better understanding of Poisson and Negative Binomial model fitting and selection. (Line: 321-324; 335-338). Also, see the S2 Table of the supplementary files.

We agree with you. Done without problem. (Line: 317-318)

Results:

We followed your recommendation throughout the Results section. We changed the dashes to "to" in all BCI values to homogenize the text.

Figures:

We corrected our figures with the help of the digital diagnostic tool Preflight Analysis and Conversion Engine (PACE), so we uploaded the files again.

---

## [Editor Report · Decision Letter 2]

23 Aug 2024

Availability of alternative prey rather than intraguild interactions determines the local abundance of two understudied and threatened small carnivore species

PONE-D-23-33252R2

Dear Dr. Santos-Moreno,

We’re pleased to inform you that your manuscript has been judged scientifically suitable for publication and will be formally accepted for publication once it meets all outstanding technical requirements.

Kind regards,

Luca Nelli, PhD

Academic Editor

PLOS ONE
---

## [Editor Report · Acceptance letter]

3 Sep 2024

PONE-D-23-33252R2 

PLOS ONE

Dear Dr. Santos-Moreno, 

I'm pleased to inform you that your manuscript has been deemed suitable for publication in PLOS ONE. Congratulations! Your manuscript is now being handed over to our production team.

Kind regards, 

on behalf of

Dr. Luca Nelli 

Academic Editor

PLOS ONE